# Graph-Directed Approach for Downselecting Toxins for Experimental Structure Determination

**DOI:** 10.3390/md18050256

**Published:** 2020-05-14

**Authors:** Rachael A. Mansbach, Srirupa Chakraborty, Timothy Travers, S. Gnanakaran

**Affiliations:** 1Theoretical Biology and Biophysics, Los Alamos National Laboratory, Los Alamos, NM 87545, USA; mansbach@lanl.gov (R.A.M.); srirupac@lanl.gov (S.C.); tstravers@lanl.gov (T.T.); 2Center for Nonlinear Studies, Los Alamos National Laboratory, Los Alamos, NM 87545, USA

**Keywords:** conotoxins, protein structure determination, homology modeling, network analysis

## Abstract

Conotoxins are short, cysteine-rich peptides of great interest as novel therapeutic leads and of great concern as lethal biological agents due to their high affinity and specificity for various receptors involved in neuromuscular transmission. Currently, of the approximately 6000 known conotoxin sequences, only about 3% have associated structural characterization, which leads to a bottleneck in rapid high-throughput screening (HTS) for identification of potential leads or threats. In this work, we combine a graph-based approach with homology modeling to expand the library of conotoxin structures and to identify those conotoxin sequences that are of the greatest value for experimental structural characterization. The latter would allow for the rapid expansion of the known structural space for generating high quality template-based models. Our approach generalizes to other evolutionarily-related, short, cysteine-rich venoms of interest. Overall, we present and validate an approach for venom structure modeling and experimental guidance and employ it to produce a 290%-larger library of approximate conotoxin structures for HTS. We also provide a set of ranked conotoxin sequences for experimental structure determination to further expand this library.

## 1. Introduction

Toxins have for a long time been considered a rich natural source of therapeutic leads because of their high specificity and binding affinity for various receptors involved in different biological pathways [1,2]. The drug ziconotide, for example, is a potent analgesic derived from a toxin produced by the aquatic cone snail species *Conus magus* [3]. The on-average smaller size of toxins–typically <100 amino acids along with a sizeable proportion <30 amino acids long [4]—means they can be employed with relative ease in in silico high-throughput screening (HTS) to rationally identify candidates for initial scaffolds interacting with a particular receptor of interest. Although traditional HTS has focused largely on small molecules, the dwindling rate at which such drugs come to market has led to a need to search for other spaces in which to identify ligands for binding with receptors of interest. Natural products, in general, are expected to be a good source of potential therapeutic candidates, and the computational advancements in various HTS strategies make it possible to apply approaches such as docking to more than just small molecules [5,6,7]. Short toxins in particular are of interest because of their pre-existing strong affinities for protein receptors, and software has been developed for in silico screening of them [8]. In one recent study of note, for example, the authors employed a docking approach to identify α-conotoxin BuIA, produced by species *Conus bullatus*, as a competitive agonist for the lysophosphatidic acid receptor 6, a G-protein coupled receptor involved in the development of several cancers [9]. Aside from their therapeutic value, toxins such as these also pose a threat to biosecurity. Rapid advances in synthetic biology have created challenges in determining the health risks posed by natural toxins or modified toxins with even higher pathogenicity [10]. Thus, in tandem with HTS for therapeutic design, it is necessary to simultaneously perform HTS for threat identification. However, HTS approaches for any application are limited by the necessity of possessing a library of at least low-resolution structures of potential toxin candidates: more structures mean a larger search space and hence a higher likelihood of identifying good initial leads [11]. Thus, structural characterization stands as a rate-limiting step for HTS for both therapeutic design and toxin threat characterization, as identified sequences often far outnumber determined structures. Indeed, only about 3% of sequences isolated from cone snail venom have corresponding experimentally-determined structures [12].

If the structures of proteins could be rapidly predicted strictly from their sequences, structural determination would not be a bottleneck; however, structure prediction from sequence still remains a challenging proposition [13]. Ab initio or de novo modeling approaches for obtaining protein structure predictions by modeling essential folding physics are prohibitively expensive except for small proteins of about 20–62 residues in size [14,15,16,17]. Even for proteins short enough to be de novo modeled in isolation, such techniques can become expensive if a large number of different structures are desired. Structure prediction for a query sequence becomes more tractable when experimentally-resolved structures are available for evolutionarily related sequences: this is referred to as homology modeling [18]. For typical proteins (at least 100 amino acids long), a useful rule of thumb for building a homology model of a protein with unknown structure using a structurally characterized protein as the template is that both proteins should share at least 25% sequence identity [19,20]; however, this does not apply to shorter peptides, since the shorter the peptide, the more likely a sequence identity of 25% is to have arisen due to random chance [21,22]. To apply the homology modeling framework for shorter peptides, a reasonable heuristic instead becomes that the alignment length and percent identity fall above the phenomenological curve introduced by Rost [23] (see Equation (Equation 1) and Appendix B
Figure A1). The relative steepness of the Rost curve for alignment lengths of less than fifty amino acids provides an illustration of why, for peptides of such lengths, it is important to use the actual functional form, rather than a static cutoff, to assess whether a pairwise alignment contains sufficient information for homology modeling.

In this article, we employ a simple sequence graph-based algorithm for homology modeling of related toxin sequences and use the resultant graph to suggest a set of sequences of interest for experimental structural characterization. Network or graph theory is, broadly speaking, the study of the ways in which different types of objects may be related to one another [24]. In essence, the basis of such approaches is to consider a set of related objects as nodes in a graph and the relationships between them as edges between those nodes. This is a versatile framework of thinking that has been profitably employed in diverse domains in the biological sciences [25,26]: e.g., biological systems modeling, in which nodes may be, for example, proteins, and edges may be chemical reactions [27,28]; elucidating allosteric pathways in proteins, in which nodes can be defined as residues, and edges as inter-residue distances [29]; antibody design, in which nodes may be glycans and edges may be the inverses of the spatial distance between them such that gaps in the network represent likely locations for antibodies to dock on a glycosylated protein [30]; or biased sampling of cluster formation, in which the nodes may be particles and edges may be physical contact between them [31]. Thus, for large datasets, network analysis techniques can help wean out salient global attributes from an otherwise confounding plethora of features. Because of the subject’s long history, casting biological problems in the form of a graph allows for the immediate application of well-verified techniques.

Since sequences of proteins may be related in many different ways, including simple amino acid identity and evolutionary relationships, it is no surprise that graph theory has a long and storied history of usage for sequence-grouping tasks such as homology detection [32], structure prediction [33,34,35], protein family identification [36,37], and even direct homology modeling [35]. For large heterogeneous databases, it can be challenging to identify homologs and a number of sophisticated algorithms have been developed for such purposes; we instead focus on the problem of homology modeling a set of cysteine-rich toxins known to be evolutionarily related. In our approach, we employ the number and placement of cysteines within a sequence as a rough initial estimate of functional and structural relatedness.

In the following sections, we present our graph-based approach and employ it to construct sequence graphs and identify good libraries of templates for homology modeling. We demonstrate that, despite the known relationship between the conotoxins, these libraries improve outcomes for structure homology modeling over using a 25% flat cutoff (plus 5% padding). We use our sequence graphs to construct a set of tables indicating sequences in which experimental structural characterization is predicted to be most valuable in creating a broad structure library by using homology modeling. Finally, we employ the graphs and libraries as part of a homology modeling procedure that results in a library of low-resolution structures for the conotoxins that will be of use in future high-throughput studies.

## 2. Results

We initialize the algorithm (see Figure 1 for a schematic illustration of the procedure) by separating a set of over 2000 known conotoxin sequences into databases containing four, six, eight, and ten cysteines, respectively. For each database, we construct graphs of sequences (cf. Figure 2) in which an edge between two nodes (i.e., sequences) represents a pairwise alignment that is of sufficient length and percent identity to fall into the safe homology modeling zone above the Rost curve (cf. Equation (Equation 1) and Figure A1). Some portions of the sequences have known structures, such that the corresponding nodes are annotated with the relevant PDB ID(s). We employ the graphs thus generated to iteratively add nodes with structures to a library of templates for homology modeling. We term this set of sequences $\left\{L_{\mathrm{ex}}\right\}$, the set of existing structural library templates. Nodes are added to $\left\{L_{\mathrm{ex}}\right\}$ in a greedy manner, in order of highest node degree, such that the resulting library will contain enough templates to homology model the maximum number of non-structurally-characterized sequences possible but with minimal sequence overlap and retaining a number of non-library structures for quality assessment. Since this is approximately the vertex-covering problem of a graph, we cannot find a globally optimal solution, as that problem is NP-complete [38]. We halt the procedure once either we have no further nodes with structures to add or there are no remaining sequences in a given connected component of the graph that are not connected to at least one library template sequence, such that all sequences in that component may be structurally characterized by homology modeling. We refer to the set of sequences that may be homology modeled based on set $\left\{L_{\mathrm{ex}}\right\}$ as set $\{C\left(L_{\mathrm{ex}}\}\right)$ that are covered by $\left\{L_{\mathrm{ex}}\right\}$. We next perform a similar procedure—but without the constraint of structure annotation—on the nodes absent $\left\{L_{\mathrm{ex}}\right\}$ to identify the sets $\left\{L_{\mathrm{proj}}\right\}$ that are of interest for experimental structural characterization such that they cover the remaining set $\{C\left(L_{\mathrm{proj}}\}\right)$. Note that it is possible for nodes to belong to both $\{C\left(L_{\mathrm{ex}}\}\right)$ and $\left\{L_{\mathrm{proj}}\right\}$, and indeed, a small number do.

In Figure 2, we present the sequence graphs for sets of conotoxin sequences with four, six, eight, and ten cysteines, respectively. We specifically display $\left\{L_{\mathrm{proj}}\right\}$ (in green), the experimental structural characterization of which would lead to coverage by homology modeling of the set $\left\{C\left(L_{\mathrm{proj}}\right)\right\}$ (in magenta) that comprises sequences with no characterized structure and not covered by set $\left\{L_{\mathrm{ex}}\right\}$. We also show the set $\left\{L_{\mathrm{ex}}\right\}$ (in orange), which we employ to predict structures for the set $\left\{C\left(L_{\mathrm{ex}}\right)\right\}$ (in blue) by homology modeling. In Figure A2, we present the same sequence graphs, but we color the nodes by relative sequence length instead of set occupation. A significant proportion of isolated sequences (nodes with no connections that therefore cannot be homology modeled) are relatively short (cf. the ring of small red nodes in Figure A2A and to a lesser extent in Figure A2B), which demonstrates that a high proportion of isolated nodes may be characterized well through rapid ab initio modeling rather than needing full experimental characterization, particularly for the four and six cysteine sequences.

These figures are a graphical illustration of the sequence space of conotoxins that may be used as a guide to experiment or to produce low-resolution structures for initial HTS. Specifically, out of the 801 sequences with four cysteines, 61 (7.6% of total) currently have experimentally-resolved structures. The graph-based approach selected 49 (6.1% of total) of these structures as comprising the four cysteine template library (set $\left\{L_{\mathrm{ex}}\right\}$; blue circles in Figure 2A, while the 12 unselected structures are represented in black), which allowed for homology modeling of 143 (17.9% of total) sequences (set $\left\{C\left(L_{\mathrm{ex}}\right)\right\}$; orange circles in Figure 2A). This corresponds to an increase of over 230% (143/61) for the number of structures employable for HTS over the original 61. The graph-based approach indicated that seven sequences from $\left\{C\left(L_{\mathrm{ex}}\right)\right\}$ (hybrid green/blue circles in the graph) and a further 74 sequences (81 overall—10.1% of total) would need to be characterized experimentally to allow for homology modeling of the remaining 151 (18.9% of total). Of the 453 sequences assigned to $\left\{L_{\mathrm{proj}}\right\}$, 372 (82.1%) are isolated nodes with no edges; of these, 298 (80.1%) are shorter than 20 amino acids, and 353 (94.9%) are shorter than 30 amino acids in length and thus many of this remainder may be rapidly modeled using ab initio techniques.

For conotoxins with six cysteines (cf. Figure 2B), 44 (4.0% of total) out of the 1113 sequences currently have experimentally-resolved structures. The graph-based approach selected 30 (2.7% of total) of these structures as comprising the six cysteine template library, which allowed for homology modeling of 148 (13.3% of total) sequences. This corresponds to an increase of over 330% (148/44) for the number of structures employable for HTS over the original 44. The graph-based approach indicated that seven sequences from $\left\{C\left(L_{\mathrm{ex}}\right)\right\}$ and a further 180 sequences (187 overall–16.8% of total) would need to be characterized experimentally to allow for homology modeling of the remaining 509 (45.7% of total). Of the 419 sequences assigned to $\left\{L_{\mathrm{proj}}\right\}$, 239 (57.0%) are isolated nodes; of these, 86 (36.0%) are shorter than 20 amino acids, and 163 (68.2%) are shorter than 30 amino acids in length.

For conotoxins with eight cysteines (cf. Figure 2C), 2 (1.1% of total) out of the 190 sequences currently have experimentally-resolved structures. These two structures were selected as comprising the entire template library here, which allowed for homology modeling of a 17 (8.9% of total) sequences. This corresponds to an increase of 850% (17/2) for the number of structures employable for HTS over the original two. The graph-based approach indicated that one sequence from $\left\{C\left(L_{\mathrm{ex}}\right)\right\}$ and a further 29 (30 overall—15.8% of total) would need to be characterized experimentally to allow for homology modeling of the remaining 101 (53.1% of total). Of the 71 sequences assigned to $\left\{L_{\mathrm{proj}}\right\}$ sequences, 41 (57.7%) are isolated nodes, but of these, only 5 (12.2%) are shorter than 30 amino acids in length, leaving a good proportion that would likely require experimental characterization to solve their structures. Finally, there are no known structures corresponding to ten cysteine sequences so there is no current coverage (cf. Figure 2D). The graph-based approach indicated that a further 9 of the total 53 sequences (17.0%) would have to be characterized to allow for homology modeling of the remaining 34 (64.2%). Of the 19 sequences assigned to $\left\{L_{\mathrm{proj}}\right\}$ (52.6%), 10 are isolated nodes, and of these, none are shorter than 30 amino acids in length.

For characterization of the remaining structures, focusing efforts on the nodes of the highest degree in the graphs is disproportionately rewarding, since a greater degree in the graph corresponds to the ability to cover a greater number of sequences. In Table 1, Table 2, Table 3 and Table 4, we present and rank the set of conotoxins that are of greatest interest for experimental characterization, as availability of experimental structures for these sequences (belonging to set $\left\{L_{\mathrm{proj}}\right\}$) would allow homology modeling of the remainder of the sequences (belonging to set $\{C\left(L_{\mathrm{proj}}\}\right)$). Thus, we suggest that experimental structural resolution begin with those sequences listed at the top of their respective tables and work downwards in order to most rapidly and efficiently structurally characterize the sequence space of the conotoxins.

In Figure 3, we assess the quality of the template libraries (Table A1, Table A2 and Table A3) constructed using the graph-based approach employing the Rost cutoff, and compared with a set of template libraries based on a static 25% rule-of-thumb cutoff. We perform two assessments: an “in-library” assessment, and an “out-of-library” assessment. In the in-library assessment (Figure 3A,B), we construct homology models for each structure in the library using the rest of the library as potential templates and compute the backbone root-mean-square deviation (RMSD) between each modeled structure and the corresponding experimental structure. Compared to the static cutoff, the Rost cutoff produces modeled structures with lower RMSD: the mean RMSD employing the Rost cutoff has a downwards shift from 4.0±0.7 Å to 1.5±0.2 Å for the four cysteine library and from 3.8±0.6 Å to 2.1±0.2 Å for the six cysteine library. In the out-of-library assessment (Figure 3C,D), we construct homology models for the known structures that were not employed as part of the template libraries (Figure 2, black nodes), and compute the backbone RMSD between each modeled structure and the corresponding experimental structure. The mean RMSD has a downward shift from 1.7±0.1 Å to 1.0±0.2 Å for the four cysteine library and from 1.82±0.09 Å to 1.4±0.1 Å for the six cysteine library. The relatively low RMSD for the in-library assessment demonstrates that the template libraries cover a substantial proportion of the known sequence space, while the relatively low RMSD for the out-of-library assessment demonstrates the utility of the homology-modeled structures as part of HTS [41]. Despite the known relationships among the toxins, there is a statistically significant improvement (downwards shift in the distribution, two-tailed Kolmogorov–Smirnov test with p<0.05) for both in- and out-of-library structures when using the Rost cutoff as compared to the 25% cutoff.

Finally, in Supporting File “finalmodels.zip”, we attach the set of structures computed by homology modeling (see Figure 4 for a schematic of the procedure), corresponding to sequences in the set $\left\{C\left(L_{\mathrm{ex}}\right)\right\}$, with the four, six, and eight cysteine library structures used as templates. Because we divided the sequences into subsets based on the number of cysteines in a sequence, we are able to use the cysteines aligned as an additional criterion during the homology modeling procedure. The average PROCHECK G-factor, which is a log-odds score based on the likelihood of observing the given distributions of ϕ-ψ and $\chi_{1}$-$\chi_{2}$ angles in proteins, is 0.086±0.005 for the reported four cysteine models, −0.103±0.007 for the reported six cysteine models, and −0.2±0.1 for the report eight cysteine models. Since this score is not a relative measure and values above −0.5 are generally considered acceptable, this provides evidence that the structures we have computed are physically reasonable. We further assess the quality of the homology modeling protocol by using it to model each structure in the library with templates selected from other structures in that library. The distribution of root-mean-square deviation (RMSD) values of the top three models based on our ranking criteria (see Materials and Methods for details) compared with each experimental structure is shown in Figure 5A,B. We see that our method performs well [41]: the average RMSD in the four cysteine architecture is 2.00±0.09 Å with at least 80% of the models having less than 3 Å RMSD, and the average RMSD in the six cysteine architecture is 2.3±0.2 Å with 75% of the models having less than 3 Å RMSD. Most of the higher RMSD values are contributed by the flexible loops and coils. When we look at the RMSD distribution after rejecting those atoms that cannot be structurally aligned, as in case of loops and coils, the distributions improve significantly (Figure A3), with a mean of 1.55±0.09 Å for the four cysteine architecture and a mean of 1.2±0.1 Å for the six cysteine architecture, with 100% of the models for both archtiectures having less than 3.5 Å deviations. A second test for validating our method was performed by checking the distribution of native contacts in the modeled structures (Figure 5C,D). Two pairs of residues were defined to have a native contact if the distance between the Cα atoms in the native experimental structure was less than 8 Å, and the pair was at least four residues apart (C $\alpha^{i}$– C $\alpha^{i+4}$). At least 60% of the native structures were captured in our models, with the distribution means of 80%±1% and 81%±1% for the four and six cysteine architectures, respectively.

## 3. Discussion

By employing a conceptually simple heuristic approach that may also be used for analysis of other short, disulfide-rich, evolutionarily-related peptides, we have constructed a set of sequence graphs that allowed us to rank non-isolated sequences without corresponding characterized structures in an order that would allow for the most rapid expansion of the conotoxin structure library. We constructed template libraries for homology modeling of conotoxins based on the number of cysteines contained in the sequence. We demonstrated that libraries constructed to account for the shorter lengths of the conotoxins produce homology models that are more accurate than libraries constructed with a static 25% cutoff. Currently, sufficient information is not available to homology model any sequences containing more than eight cysteines, as experimental characterization has focused preferentially on the shorter conotoxins. We employed our libraries to predict a set of structures from sequence using homology modeling, allowing us to expand the library of conotoxin structures usable for HTS by about 290% overall, although a number of sequences remain without any associated structural predictions. We assessed the quality of these structures through standard techniques to demonstrate they are expected to be reasonably accurate and therefore may be employed for high-throughput screening of conotoxins as novel therapeutics for new receptor targets. We note that of those sequences that were isolated in our graphs—that is, had no edges—80% of those containing four cysteines and 36% of those containing six cysteines were under 20 amino acids long, marking them as good candidates for a high-throughput ab initio modeling procedure, rather than necessarily for experimental characterization, as they will likely be tractable but will not contain any information about other sequences.

For experimentalists interested in investigating the conotoxins we suggest in the table, we note that the preponderance of peptide structures that we employed for the structure libraries were determined with solution nuclear magnetic resonance (NMR). Maintenance of the tertiary fold of conopeptides, particularly after their secretion into the extracellular environment, depends more on its disulfide bonds and less on the presence of a hydrophobic core [44]. As such, compression artifacts on the free/unbound structures of these disulfide-rich peptides can arise due to crystal packing forces in X-ray studies, which has been observed in the literature [45]. It was also shown recently that solution NMR structures of disulfide-rich peptides based primarily on nuclear Overhauser effects (NOEs) are comparable to a crystallographic resolution of around 2.5 Å, while refinement with residual dipolar couplings (RDCs) can improve this to around 1–1.5 Åresolution [46]. Solution NMR studies, therefore, in addition to providing ensembles that inform on the conformational dynamics, can yield highly appropriate and cost-effective structures compared to those obtained from X-ray or cryo-EM [47].

One important point about short, disulfide-rich peptides that we have not addressed in this work is the existence of so-called “disulfide isomers.” Under certain environmental conditions, there is experimental evidence suggesting that some toxins do not exist as a single set of “native” structures but as a heterogeneous—perhaps metastable—ensemble populated with strikingly different conformations corresponding to differing patterns of cysteine connectivity [48]. Although many conotoxins do have a single thermodynamically stable native state as dictated by their sequence similarity, [49,50] or can be stabilized in one through the use of dicarba or diselenide bonds [51,52], these disulfide isomers, if kinetically or thermodynamically controlled, may be employed to expand the library of structures for HTS. This represents an important area of future work.

Although we do not directly address the question of structure–function relationships in this work, the graph-based method does suggest an interesting route forward. It has been shown that in addition to strict structural contributions, conotoxin binding is also heavily related to surface electrostatics [53], and, indeed, this concept was formalized (as “Protein Surface Topography”) and employed by Kasheverov et al. [54] to design stronger-binding α-conotoxin mutants. Another intriguing line of further study, therefore, might be to construct a graph on the basis of the Protein Surface Topography similarities for those structures we have modeled in this work and compare and contrast between that and sequence similarity.

## 4. Materials and Methods

### 4.1. Data Acquisition and Curation

For use in construction of the template libraries, we employed a set of 142 conotoxin structures downloaded from the PDB [55], which we found by searching “conotoxin” on the PDB. We manually removed several false positives, such as a crystal structure of the acetylcholine-binding protein that was identified due to the title of the associated paper. We also manually removed several sequences that were identical to natural conotoxin sequences but modified by the replacement of disulfide bonds with dicarba bonds. We did not remove redundant sequences consisting of multiple characterization methods and, in a few cases, structural isomers resulting from different disulfide-bond connections. In future, further work will be done to properly assess the likelihood of multiple stable or metastable states, but we do not address this consideration further here.

For use in the analysis detailed in this article, we downloaded a set of 6255 peptide sequences from the Conoserver [40] using the “Tools > Download Conoserver’s Data” command. We retained only sequences containing four, six, eight, or ten cysteines. We removed anything with the word “precursor” or “patent” in the name, as precursor sequences contain, in addition to the mature peptide sequence that folds into the toxin, a signal sequence and N- and C-terminal pro-regions that are cleaved in the endoplasmic reticulum and Golgi apparatus [56]. A manual inspection of sequences labeled “patent” revealed that many were insufficiently characterized—for example, they noted only the cysteine pattern or they mixed precursor and mature toxin sequences with no indication. We also added to the sequence list any sequence that corresponded to one of the PDB structures that was not already contained in the list. Once the set of all sequences was finalized, we split it into four subsets corresponding to the number of cysteines contained. In the end we retained for analysis a total of 801 unique sequences containing four cysteines, 1113 unique sequences containing six cysteines, 190 unique sequences containing eight cysteines, and 53 unique sequences containing ten cysteines.

### 4.2. Details of Library Template Selection Procedure

For each subset of sequences corresponding to a different number of contained cysteines, we created an alignment graph as follows. For every sequence, we computed a pairwise alignment with every other sequence, using the “PairwiseAligner” class in the “Align” module of the Biopython package [57], in global mode, with a gap-open penalty of -10 and a gap-extend penalty of −0.5. Employing the networkx Python package [58], we constructed a graph in which nodes represented sequences and we placed an edge between two nodes whenever the percent identity of the highest-percentage pairwise alignment of the two corresponding sequences was greater than [23],
(1)$$p_{\mathrm{rost}}=n+480L^{-0.32\left(1+\exp\left(-\frac{L}{1000}\right)\right)},$$
where *L* is the length of the alignment in numbers of amino acid residues and we set n=5 (%).

We constructed two different template libraries for each subset of sequences, one from the pairwise alignment graph and one from a static 25%-identity cutoff (with n=5 %). When creating the graph-based libraries (see also Figure 1), we first identified all connected components in the graph. For each connected component, we chose first the sequence with the highest node degree (number of distinct edges) that corresponded to a structure in the set of 142 structures downloaded from the PDB, added it to the library, and removed that sequence and all sequences it shared an edge with from the graph. We continued this procedure until one of two criteria was satisfied: (i) there were no longer any sequences in the connected component with corresponding structures or (ii) there were no longer any sequences in the connected component without corresponding structures. These criteria corresponded to the following two situations, respectively: (i) there were no other structures available for inclusion in the library or (ii) the entire connected component was able to be homology modeled based on the structures included in the library up until that point. For construction of the static sequence-identity cutoff library, sequences within each data set were clustered and a representative sequence from each cluster chosen by using the “sequence_db.filter” command of MODELLER version 9.20 (UCSF, San Francisco, CA, USA) [59,60], which groups sequences together if their sequence identity is greater than a specified cutoff value. The set of cluster representatives became a library of structures in which between any pair the sequence identity was less than the specified cutoff value.

For computation of the homology modeled structures based on the library templates that were used to assess and compare the quality of the two libraries, we used the “align2d” command followed by the “automodel” procedure from MODELLER 9.20 with default parameters. We computed five models for each sequence from each template (except for itself, in the case of library structures being modeled based on other library structures). The best homology model was chosen as the one with the lowest backbone RMSD to the known or experimentally-resolved structure, using the “align” command in PyMOL [61] that superimposes two structures via a structure superposition that is constrained by a prior sequence alignment.

### 4.3. Homology Modeling Criteria

After assessing the quality of the template libraries, we used them to construct via homology modeling a database of structures for those conotoxin sequences (set $\left\{C\left(L_{\mathrm{ex}}\right)\right\}$ shown in blue in Figure 2 and Figure A2) that are covered by those libraries (set $\left\{L_{\mathrm{ex}}\right\}$ given in orange in Figure 2 and Figure A2). The pipeline employed for building these homology-modeled structures is detailed here. A schematic of this pipeline is given in Figure 4. The four cysteine (4C) subset included 143 such sequences for which structures were computed by homology modeling from 49 library structure templates, while the six cysteine (6C) subset included 148 sequences for which structures were computed by homology modeling from 30 library structure templates. There was only one existing non-isolated library structure template having eight cysteine (8C) architecture, and 17 sequences were modeled from it, while no ten cysteine (10C) structures could be modeled, since to date there are no structures of conotoxins containing 10 cysteines deposited in the PDB.

Alignment of each of the subject sequences was performed with those sequences that have a structure present in the template library using BLAST [42]. BLOSUM62 substitution matrix [62] was used with a gap-open penalty of −10 and a gap-extend penalty of −0.5. For each sequence, structures were considered possible templates if they fulfilled the following criteria: (i) sequence identity of ≥70%; (ii) ≥70% of sequence length covered; (iii) E-value $\leq 1\times 10^{-5}$. Additionally, we constrained the cysteines in the sequence to be aligned in the following manner. If there was a one-position shift in the sequence alignment that would allow the cysteines to align, the gap penalties at that position were removed to enforce cysteine alignment. If a greater than one-position shift would be required to allow the cysteines to align, such a template was not considered.

Structural homology modeling was performed using the MODELLER version 9.20 package [59,60]. Multiple templates were used to aid in the modeling process for those subjects where more than one sequence satisfied the above-mentioned criteria. The models were further relaxed by several steps of conjugate gradients and molecular dynamics with simulated annealing as recommended in the thorough Variable Target Function Method (VTFM) optimization of MODELLER [63]. Due to the alignment of cysteines from the template structures, the disulfide bonds could be constrained by patches. Ten such models were generated for each subject sequence. Subjects 107 and 110 from 4C architecture and subject 2 from 6C architecture did not correspond to any templates that satisfied all of our above criteria. Nevertheless, we modeled these sequences based on the best sequence match.

We selected three top models for each subject based on the Discrete Optimized Protein Energy (DOPE) score [64] and the PROCHECK G-factor [65]. DOPE, a typical criterion for assessing the quality of a modeled structure, is an atomistic distance-dependent statistical potential calculated from a large set of refined high resolution PDB structures. The PROCHECK G-factor is a log-odds score based on observed distributions of the ϕ-ψ, and $\chi_{1}$-$\chi_{2}$ values measuring whether the model is physically reasonable or if it contains unusual stereochemical configurations. In this study, we normalized the DOPE and G-factor scores and used a combined product of probabilities to rank and sort the structures. The top three models selected for each subject are reported in Appendix A “finalmodels.zip”, along with their DOPE, G-factor, MODELLER optimization function value (molpdf), GA341 scores [66], and the Ramachandran plots for each of these models. All RMSD calculations were performed with Pymol [61]. There is only one available non-isolated structure in the 8C extant library. This was used to model all 17 subject sequences. The best three models for each sequence along with their assessment scores are reported in the database.

### 4.4. Quantification and Statistical Analysis

We use the Kolmogorov–Smirnov two-tailed test as implemented in the SciPy package [67] and referred to by the “ks2samp” command to assess whether we may reject the null hypothesis of the RMSDs of experimental structures from homology models based on different template libraries being drawn from the same distribution. We employ a significance level of p=0.05, meaning that we reject the null hypothesis if the KS statistic *D* returned by the test is such that $D>\alpha \sqrt{\frac{n+m}{nm}}$, where α=1.224 for a significance level of p=0.05, and *n* and *m* are the number of samples in each set, respectively. The analysis is referred to in Section 2.

## 5. Conclusions

Overall, the work in this article presents a rational graph-based algorithm that we employ to expand the repertoire of known conotoxin structures for application in a high-throughput manner as part of the early stages of drug design. We expect that the libraries, the expanded set of structures, and the ranking of sequences in terms of the degree of connectedness to other sequences will be valuable resources improving the prospects of conotoxins as novel therapeutic leads and that our approach may be employed for initial characterization of other sets of evolutionarily-related toxins.

## Figures and Tables

**Figure 1 marinedrugs-18-00256-f001:**
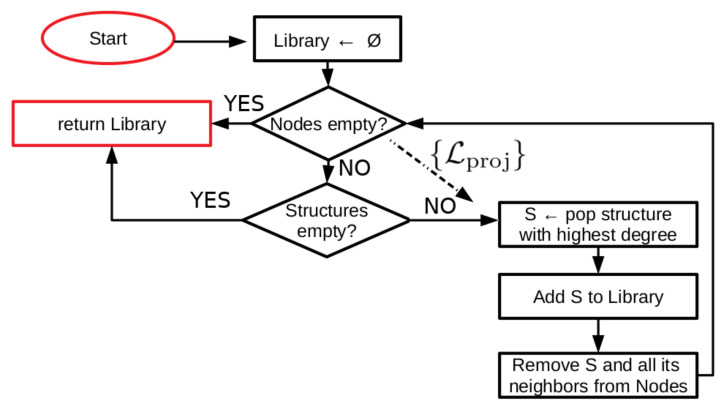
Schematic of a simple graph-based algorithm for constructing a library of structural templates for homology modeling. For each connected component in the graph of sequences, where an edge represents the ability to homology model one sequence based on another, we employ a greedy approach to find a good library of template structures that cover as much of the sequence space as possible. For computation of the sequence set $\left\{L_{\mathrm{proj}}\right\}$ of interest for experimental characterization, we skip consideration of the structures and run the algorithm on the subset with structure-associated sequences removed.

**Figure 2 marinedrugs-18-00256-f002:**
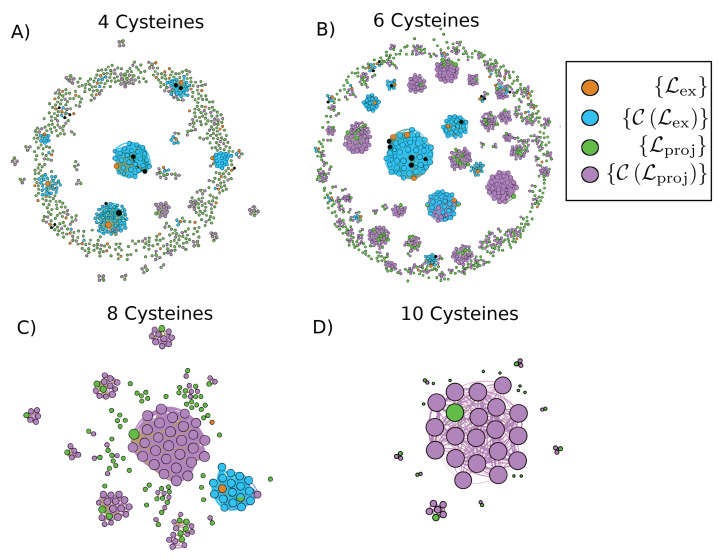
Graph of conotoxins containing (**A**) four cysteines, (**B**) six cysteines, (**C**) eight cysteines, and (**D**) ten cysteines where nodes are sequences and edges exist between sequences with pairwise alignments that have high enough length and percent identity to fall above the Rost curve with n=5% (Equation (Equation 1)). We show the set $\left\{L_{\mathrm{ex}}\right\}$ of sequences added to the template libraries in orange, the set of sequences corresponding to unselected structures in black, the set of covered sequences $\left\{C\left(L_{\mathrm{ex}}\right)\right\}$ that we homology model based on the templates included in the library in blue, and the set of projected sequences $\left\{L_{\mathrm{proj}}\right\}$ in green in which structures are in need of characterization in order that the rest of the sequences $\left\{C\left(L_{\mathrm{proj}}\right)\right\}$ in magenta may be homology modeled based on some template. Nodes belonging to both $\left\{C\left(L_{\mathrm{ex}}\right)\right\}$ and $\left\{L_{\mathrm{proj}}\right\}$ are displayed as half green, half blue. The sizes of the nodes correspond to their degree; that is, the number of other sequences that they can be modeled based on or used to model. Node locations and edge lengths were chosen for ease of visualization of separate connected components. Visualization of the graphs was produced with Gephi 0.9.2 [39].

**Figure 3 marinedrugs-18-00256-f003:**
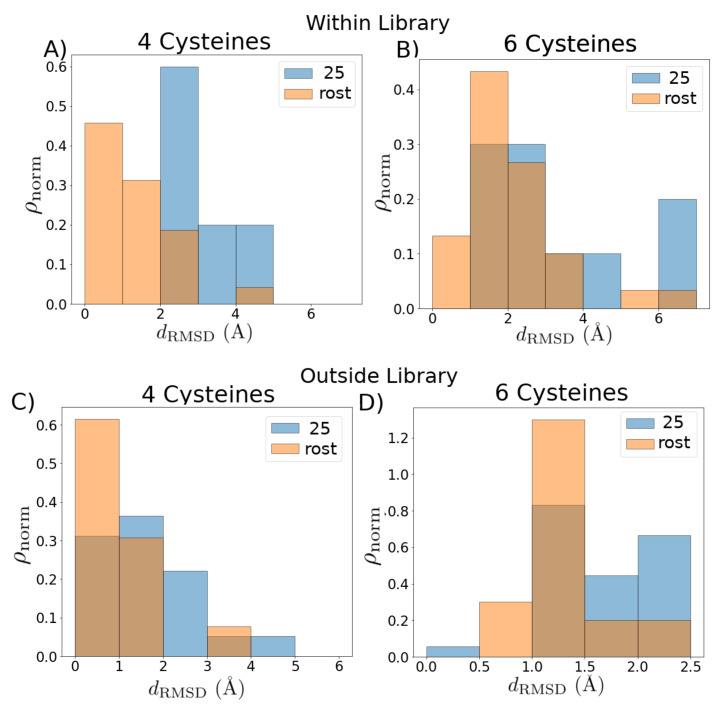
Quality of graph-based template library selection criteria. Comparison of root-mean-square deviation (RMSD) distributions from experimental structures for (**A**,**B**) structures within the libraries, with each structure modeled by selecting from all other templates within the given library (“in-library” assessment), and (**C**,**D**) structures outside the libraries modeled by selecting from all templates within the given library (“out-of-library” assessment). For each homology modeled structure, we choose the best fit to the experiment. The distributions produced by the simple 25% cutoff libraries are shown in blue; the distributions produced by using the graph-based algorithm are shown in orange. Distributionis are transparent for ease of viewing.

**Figure 4 marinedrugs-18-00256-f004:**
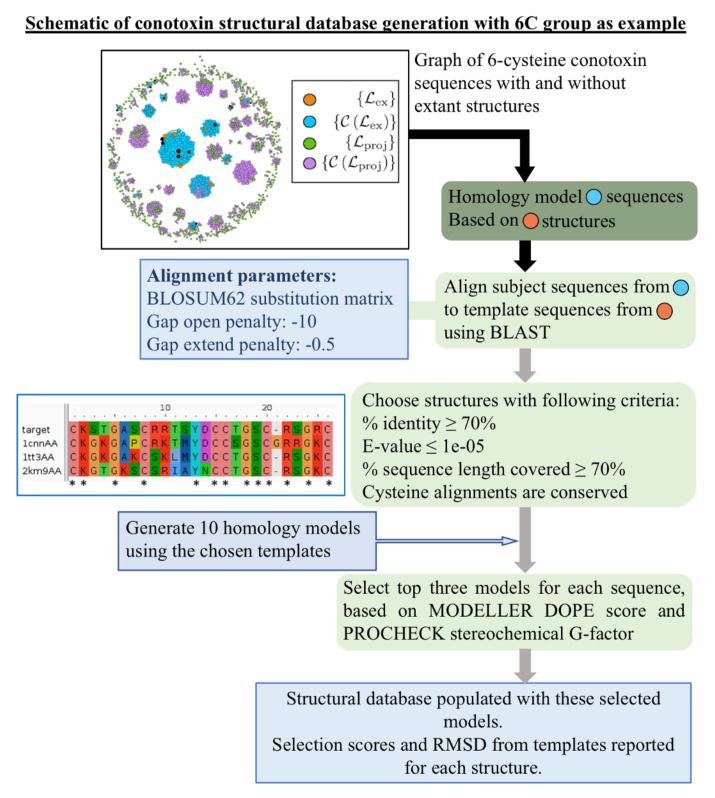
Schematic of procedure for producing homology modeled structures from library templates for conotoxin sequences with unknown structure lying in the set $\left\{C\left(L_{\mathrm{ex}}\right)\right\}$. We employ a BLAST alignment procedure and specifically force the cysteines to align to further refine the templates that were originally chosen for inclusion using the graph-based Rost criterion. Graph inset of the eight cysteine graph is an example. The inset consisting of an example alignment input figure was created using the alignment obtained from BLAST [42] and visualized with Aliview [43].

**Figure 5 marinedrugs-18-00256-f005:**
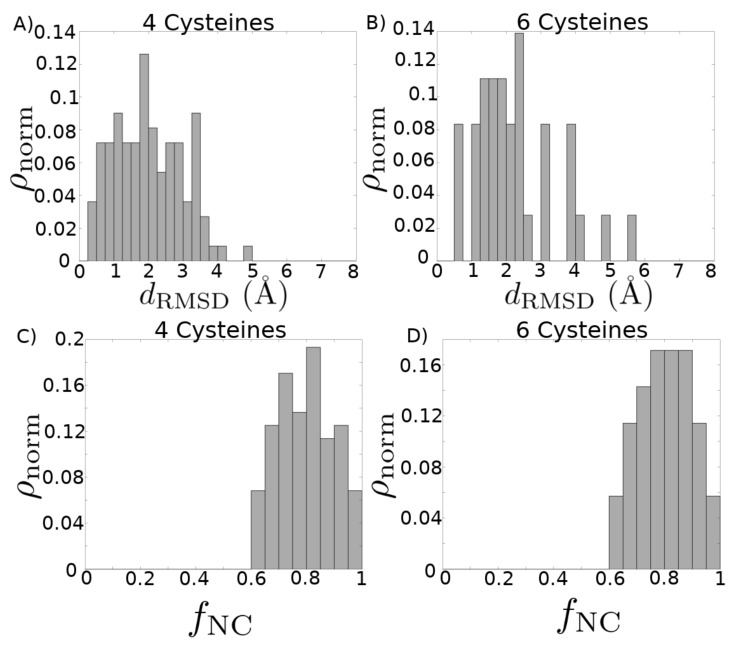
Quality of modeling criteria. (**A**,**B**) Distribution of root-mean-square deviation (RMSD) for homology models compared with their corresponding experimental structures, without prior removal of any structural alignment outliers. Each experimental structure present in the library was modeled by selecting from all other templates in the library. The top three models for each structure based on combined MODELLER DOPE and PROCHECK G-FACTOR scores are considered here. (**A**) Distribution mean = 2.00 Å, standard deviation = 0.97 Å. (**B**) Distribution mean = 2.25 Å, standard deviation = 1.20 Å. (**C**,**D**) Distribution of fraction of native contacts present in each of the homology modeled structures, with respect to the experimental structure. Each experimental structure present in the library was modeled by selecting from all other templates in the library. The top three models for each structure based on combined MODELLER DOPE and PROCHECK G-FACTOR scores are considered here. (**C**) Distribution mean = 0.797, standard deviation = 0.108. (**D**) Distribution mean = 0.805, standard deviation = 0.097.

**Table 1 marinedrugs-18-00256-t001:** List of sequences containing four cysteines in order of interest for experimental characterization, based on the degree (sequence coverage) in alignment graphs (cf. Figure 2). Name or names of sequences are taken from the Conoserver database [40]. Multiple names for the same sequence indicate the same sequence is produced by different species or has different post-translational modifications. Node degree corresponds to the number of sequences with pairwise alignments that are long enough and have high enough percent identity to be homology modeled with the given sequence as a template. Cysteines are highlighted in red to guide the eye. We note in the fourth column the pharmacological family, although it is unknown for the majority of sequences, as it requires a separate experimental determination in most cases.

Sequence	Name(s)	Degree	Pharm. Fam.
AAKVKYSNTPEECCSNPPCFATHSEICG	Li1.28	10	Unknown
GCCSDPRCAYDHPEIC	Vc1.1[N9A]	10	alpha
GCCSNPVCHLEHSNAC	MII [L15A]	8	Unknown
AALEDADMKTEKGFLSSIVGNLGTVGNLV–
GSVCCQITNSCCPED	Pu5.7	7	Unknown
RAALEDADMKTEKGVLNAIFSNLGDLGNL–
VSSVCCKATTSCCPED	Pu5.9	6	Unknown
AGLTDADLKTEKGFLSGLLNVAGSVCCKVDTSCCSNQ	Lt5g	6	Unknown
GCCSNPVCALEHSNLC	MII [H9A]	6	Unknown
VPAEQMMEELCPDMCNRGEGEIICTCVLRRHVVSPSIR	Lt14.4	5	Unknown
TNEGPGRDPAPCCQHPIETCC	Cal5b	5	Unknown
RPECCTHPACHVSNPELCS	Mr1.8	4	Unknown
GCCSRPPCIANNPDLC	TxIA	4	alpha
SPGSTICKMACRTGNGHKYPFCNCR	Fe14.1	4	Unknown
GCCSLPPCALNNPDYC	PnIA [A10L,sTy15Y]	4	Unknown
YAAVVNRASALMAQAVLRDCCSNPPCAHNIHCA	Ec1.7	4	Unknown
NGRCCHPACGKHFSC	Ac1.1b, CnIH, R1.1, Bt1.6, Mn1.2, C4.3	4	Unknown
NGRCCHPACGKYFSC	Mn1.5	3	Unknown
GCCSRAACAGIHQELC	LtIA [A4S]	3	Unknown
GCCSNPVCHLAHSNAC	MII [E11A,L15A]	3	Unknown
GCCSHPACSGNNREYCRES	O1.3	3	Unknown
GGCCSHPVCYFNNPQMCR	Cr1.6	3	Unknown
GGGCCSHPACAANNQDYC	Gly-AnIB	3	Unknown
DGCCSSPSCSVNNPDICGG	Eb1.1, Qc1.18	3	Unknown
LDPCCREPPCASTHTDICT	Li1.4, Sa1.12	3	Unknown
NECCDNPPCKSSNPDLCDWRS	Qc1.1b, LiC22	3	Unknown
DECCSNPSCAQTHPEVC	Li1.24, Sa1.6	3	Unknown
GCCSHPACAGNNPHICS	Li1.11	3	Unknown
SFRFIPGGIKEIACHRYCAKGIASAFCNCPDKRDVVSPRI	G14.1	3	Unknown
VPPEPILEIICPGMCDEGVGKEPFCHCTKKRDAVSSRI	Vc14.4	3	Unknown
GCCSYPPCNVSYPEICG	Su1.6	2	Unknown
AANDKASVQIALTVQECCADSACSLTNPLIC	Dd1.7, Li1.21	2	Unknown
TAFGLRLCCKRHHGCHPCGRT	Cal1b	2	Unknown
AANAKLFDVGQSCCSAPLCALLYMVIC	Sa1.7	2	Unknown
TVRDACCSDPRCSGKHQDLC	Li1.16, Sa1.3	2	Unknown
NLQILCCKHTPACCT	S5.3, Eb5.5	2	Unknown
ECPPWCPTSHCNAGTC	Cl14c	2	Unknown
GIWCDPPCPKGETCRGGECSDEFNSDV	Cal14.1a	2	Unknown
GMWDECCDDPPCRQNNMEHCPAS	Lp1.7	2	Unknown
GRCCHPACGGKYFKC	CnIJ	2	Unknown
IALIATRECCANPQCWSKNC	Co1.3	2	Unknown
GCCSHPVCHARHPELC	PeIA[A7V, S9H,V10A,N11R]	2	Unknown
DGCCSDPACSVNHPDICGG	Qc1.7	2	Unknown
PPGCCNNPACVKHRCG	Bu1.2	2	Unknown
LINTRCCPGQPCCRM	Vc5.11	2	Unknown
NAAANDKASDVIPLALQGCCSNPVCHVDHPELCL	Cn1.6	2	Unknown
GCCSHPVCHARHPALC	PeIA[A7V, S9H,V10A,N11R,E14A]	2	Unknown
WDVNDCIHFCLIGVVGRSYTECHTMCT	FlfXIVB	2	Unknown
NGRCCHPACAKYFSC	Mn1.4b	2	Unknown
NGRCCHPACGGKYVKC	Ac1.2	2	Unknown
GCCSYPPCFATNSDYC	AuIA	2	alpha
DECCAIPLCAKIFPGRCP	Pc1b	1	Unknown
AANLMALLQESLCPPGCYPSCTNCRYMFP	Pu14.6	1	Unknown
GCCAIRECRLQNAAYCGGIY	Ca1.2	1	Unknown
FLTQQSPRDFAKSVMQLLHYNWIDCCNYGVSDCCI	Lv5.7	1	Unknown
APAELILETICPHMCGTGIGEPFCNCRNKRDVVSSRII	Bt14.3	1	Unknown
EIVNIIDSISDVAKQICCEITVQCCVLDEE	Vn5.5	1	Unknown
ECCEDGWCCTAAPLTAP	Vc5.7	1	Unknown
CCPGWELCCEWDDWW	Mr5.7	1	Unknown
GCCSFPACRKYRPEMCG	Su1.2	1	Unknown
DDCCPDPACRQNHPELCST	PuSG1.1	1	Unknown
APNVKDSKASGSCCDNPSCAVNNSHC	Li1.32	1	Unknown
YHECCKNPPCRNKHPDLC	Sa1.16	1	Unknown
GCCSNPACAGSNAHIC	Li1.14	1	Unknown
GCCVYPPCAVNHPDICRG	Qc1.9	1	Unknown
VMQLRYYNWIDCCFDGDCCN	Qc5.3	1	Unknown
TGCCEYPYCAENNPELCG	Co1.4	1	Unknown
SVEGVISTIKDFAVKVCCSVSLKFCCPTA	Ts5.5	1	Unknown
SCCSDSDCNANHPDMCS	Leo-A1	1	Unknown
SCCPQEFLCCLYLVK	Lp5.1	1	Unknown
RCCHPACGKNYSC	MI[del1G]	1	Unknown
QTPGCCWNPACVKNRC	EIIA	1	Unknown
QGCCSYPACAVSNPDICGG	Qc1.12	1	Unknown
PECCSDPRCNSTHPELCG	Ai1.2	1	Unknown
NIQIICCKHTPKCCT	Tx5.5	1	Unknown
NAWLTPEECCAAPACREMILEFCLAGEAFAAAL–
DGFRRLPYR	Pu1.5	1	Unknown
KVYCCLGVRDDWCCAGQIQI	Lt5i	1	Unknown
IINWCCLIFYQCCL	Sr5.7	1	Unknown
YCCHPACGKNFDC	SIA	1	alpha
GILELAKTVCCSATGISICC	Tx5.13, Tr5.3, Vr5.1	1	Unknown
GGCCSRPPCILKHPEIC	Qc1.13	1	Unknown
GCPADCPNTCDSSNKCSPGFP	Cal14a	1	Unknown
GIRGNCCMFHTCPIDYSRFYCP	Vt1.24	1	Unknown

**Table 2 marinedrugs-18-00256-t002:** List of sequences containing six cysteines in order of interest for experimental characterization, based on the degree (sequence coverage) in alignment graphs (cf. Figure 2). Name or names of sequences are taken from the Conoserver database [40]. Multiple names for the same sequence indicate the same sequence is produced by different species or has different post-translational modifications. Node degree corresponds to the number of sequences with pairwise alignments that are long enough and have a high enough percent identity to be homology modeled with the given sequence as a template. Cysteines are highlighted in red to guide the eye. We note in the fourth column the pharmacological family, although it is unknown for the majority of sequences, as it requires a separate experimental determination in most cases.

Sequence	Name(s)	Degree	Pharm. Fam.
LPPCCSLNLRLCPAPACKYKPCCKS	RIIIJ Δ6-11	29	Unknown
QKGLVPSVITTCCGYDPGTMCPPCRCTNSCPKKPKKP	S4.4	28	Unknown
QPWLVPSKITNCCGYNTMEMCPTCMCTYSCRPKKKKP	Mn4.2	27	Unknown
DDECEPPGDFCGFFKIGPPCCSGWCFLWCA	MaIr137, G6.2	20	Unknown
DCVAGGHFCGFPKIGGPCCSGWCFFVCA	Vn6.8	20	Unknown
VCREKGQGCTNTALCCPGLECEGQSQGGLCVDN	Mi010	20	Unknown
ECREQSQGCTNTSPPCCSGLRCSGQSQGGVCISN	CaHr91	17	Unknown
TVDEACNEYCEERNKNCCGRTDGEPVCAQACL	Vi6.7	15	Unknown
ECTRSGGACYSHNQCCDDFCSTATSTCV	Eb6.22	15	Unknown
GCTPPGGACGGHAHCCSQSCNILASTCNA	ABVIC	15	Unknown
TVGEECNEYCEQRNKNCCGKTNGEPVCAQACL	Tr7.4	15	Unknown
TATEECEEYCEDEEKTCCGEEDGEPVCARFCL	Ar6.24	14	Unknown
EACYNAGTFCGIKPGLCCSAICLSFVCISFDLIDVFSSP	M6.2	13	Unknown
TTEECHEYCEDQNKNCCGLTDGEPRCAGMCL	Tr7.3	13	Unknown
MTMGCTHPGGACGGHYHCCSQSCNTAANSCN	MIL3-b (partial)	12	Unknown
VPEECEESCEEEEKTCCGLENGQPFCSRICW	Ar6.28	12	Unknown
DECYPPGTFCGIKPGLCCSERCFPFVCLSLEF	Ac6.2	12	Unknown
CLDAGEVCDIFFPTCCGYCILLFCA	TxO1	11	omega
DCTPPDGACGFHYHCCSKFCITISSTCN	MIL2-a	11	Unknown
CIDGGEICDIFFPNCCSGWCIILVCA	Mr6.8	11	Unknown
TTAESWWEGECLGWSNGCTHPSDCCSNYCKGIYCDL	Mr6.16	11	Unknown
GCTHPGGACGGHHHCCSLFCNTAANACN	MIL3-f	11	Unknown
CLGSGETCWLDSSCCSFSCTNNVCF	Vn6.15	11	Unknown
CLDAGEMCDLFNSKCCSGWCIILFCA	Mr6.1	10	Unknown
SCGEEGEGCYTRPCCPGLKCIGTAHGGLCREE	Pu6.7	10	Unknown
GCLEVDYFCGIPFVNNGLCCSGNCVFVCTPQ	Pn6.7	10	Unknown
SIAGRTTTEECDEYCEDLNKNCCGLSNGEPVCATACL	Ts6.7	10	Unknown
DGCYNAGTFCGIRPGLCCSEFCFLWCITFVDS	MVIA, Cn6.1	10	delta
NCCNGGCSSKWCRDHARCC	SIIIA[del1]	9	Unknown
KTTAESWWEGECYGWWTSCSSPEQC–
CSLNCENIYCRAW	TsMEKL-03	8	Unknown
RHGCCKGPKGCSSRECRPQHCC	TIIIA	8	mu
DCGEQGQGCYTRPCCPGLHCAAGATGGGSCQP	Conotoxin-1	8	Unknown
CLAGSAPCEFHRGYTCCSGHCLIWVCA	Cal6.1d	8	Unknown
KTTAESWWEGECRTWYAPCNFPSQC–
CSEVCSSKTGRCLTW	Vn6.5	7	Unknown
CRPPGMVCGFPKPGPYCCSGWCFAVCLPV	MaIr193	7	Unknown
CTPGGEACDATTNCCFLTCNLATNKCRSPNFP	ABVIL	7	Unknown
WWEGECRGWSNGCTTNSDCCSNNCDGTFCKLW	Vn6.3	7	Unknown
WWWGGCTWWFGRCSTDSECCSNSCDQTYC–
ELYRFPSRY	Vc6.26	7	Unknown
YECYSTGTFCGINGGLCCSNLCLFFVCLTFS	CnVIA, St6.2	7	delta
CCSRDCWVCIPCCPNGSA	Lv3-IP01	7	Unknown
VCVDGGTFCGFPKIGGPCCSGWCIFVCL	Ar6.2	6	Unknown
ECIEGSEPCEVFRPYTCCSGHCIIFVCA	Cal6.1h	6	Unknown
CCSQDCWVCIPCCPN	Eu3.2	6	Unknown
CCSQDCSVCIPCCPN	Co3-IP02, Ts3-IP07, Vr3-IP08, Rt3-IP03, Ca3-IP02, Ec3-IP03	6	Unknown
CYDSGTSCNTGNQCCSGWCIFVSCL	Tx6.3	6	Unknown
CTVDSDFCDPDNHDCCSGRCIDEGGSGVCAIVPVLN	Ar6.19	6	Unknown
FPCNPGGCACRPLDSYSYTCQSPSSSTANCEGNECVS–
EADW	Cl9.4	6	Unknown
GPPCCLYGSCRPFPGCSSASCCRK	PIIIF [Y17S,N18S,L20S]	5	Unknown
DCQEKWDYCPVPFLGSRYCCDGFICPSFFCA	Da6.6, Tx6.6	5	Unknown
CCGVPNAACHPCVCNNTC	OIVA [K15N]	5	Unknown
CTPRNGYCYYRYFCCSRACNLTIKRCL	Ml6.2	5	Unknown
CCSQDCRVCIPCCPN	Ts3.1	5	Unknown
QCTPVGGSCSRHYHCCSLYCNKNIGQCLATSYP	Ar6.17	5	Unknown
CLNDGDDCDTGDDCCSGLCIFDEYFSYCDDSDP–
YYDDYDEYYY	Mi029	5	Unknown
SCGNLHESCSAHRCCPGLKCIGTAHGGLCRE	Pu6.15 (partial)	5	Unknown
VKPCSEEGQLCDPLSQNCCRGWHCVLVSCV	Da6.2	5	Unknown
FAVIFTCTPPGSHCTGHSDCCSDFCSTMSDVCQ	Co6.1	4	Unknown
WWDGECRLWSNGCRKHKECCSNHCKGIYCDIW	VeG52	4	Unknown
CTPCGPDLCCEPGTTCDTVLHHTRFGEPSCSY	Fla6.16	4	Unknown
CCGKPNAACHPCVCNGSCS	G4.1	4	Unknown
MGYILPALSQQTCCVRPWCDGACDCCVDS	Co3-D01	4	Unknown
MKLMLSALRQQECCKPSTCDGGCYHCC	Lv3-YH04	4	Unknown
SCGNLHESCSAHRCCPGLMCFTLPTPICIW	Pu6.17	4	Unknown
CIPQFDPCDMVRHTCCKGLCVLIACSKTA	Pn6.3	4	Unknown
STSCMEAGSYCGSTTRICCGYCAYFGKKCIDYPSN	SO5	4	omega
GGCTPCGPNLCCSEEFRCGTSTHHQTYGEPACLSY	Ca6.2	4	Unknown
CLGFGEACLMLYSDCCSYCVALVCL	Ep6.1	4	Unknown
CIEQFDPCEMIRHTCCVGVCFLMACI	King-Kong 1	4	Unknown
TCSPAGEVCTSKSPCCTGFLCTHIGGMCHH	LvVIA 2	4	Unknown
CTPSGGACYVASTCCSNACNLNSNKCV	M1	4	Unknown
CCGVPNAACHPCVCTGKC	PeIVA	4	alpha
ATDCIEAGNYCGPTVMKICCGFCSPFSKICMNYPQN	Ac6.5	4	Unknown
GCTPRNGACGYHSHCCSNFCHTWANVCL	LvVID	3	Unknown
DVCELPFEEGPCFAAIRVYAYNAKTGDCEQLTY–
GGCEGNGNRFATLEDCDNACARY	Cal9.1d	3	Unknown
KFCCDSNWCHISDCECCY	Tx3h	3	Unknown
GCGYLGEPCCVAPKRAYCHGDLECNSVAMCVN	Mr2	3	Unknown
ECTPPEGACNHPSHCCEDFCDRGRNRCM	At6.7	3	Unknown
WWEGDCTDWLGSCSSPSECCYDNCETYCTLW	Lt7b	3	Unknown
CRSSGSPCGVTSICCGRCYRGKCT	SVIA	3	omega
CKAESEACNIITQNCCDGKCLFFCIQIPE	Pn6.5	3	Unknown
CKSPGTPCSRTMRDCCTSCLSYSKKCR	G6.12	3	Unknown
CTPPSGYCYHPYYCCSRACNLTRKRCL	At6.2	3	Unknown
PCKTPGRKCFPHQKDCCGRACIITICP	P2a	3	Unknown
CVPYEGPCNWLTQNCCDELCVFFCL	Gm6.3	3	Unknown
SKQCCHLPACRFGCTPCCW	Mr3.4	3	Unknown
CCKYGWTCWLGCSPCGC	PnIVB	2	mu
EIILHALGTRCCSWDVCDHPSCTCC	Vr3-T05	2	Unknown
CNNRGGGCSQHPHCCSGTCNKTFGVCL	VxVIA, MgJ42	2	Unknown
CAGIGSFCGLPGLVDCCSGRCFIVCLP	Bt6.4, ErVIA	2	Unknown
CCHWNWCDHLCSCCGS	Mr3.8	2	Unknown
CCQAACSPWLCLPCC	Eu3.3, Bt3.3	2	Unknown
CIPFLHPCTFFFPDCCNSICAQFICL	VcVIC	2	Unknown
CTQSSEFCDVIDPDCCSGVCMAFFCI	Vc6.40	2	Unknown
CTVNGVVCDPGNHNCCSGSCLDDEDTPVCGIHV–
EIQHVHMLS	Pu6.23	2	Unknown
CCDDSECDYSCWPCCMF	Gm3-WP04	2	Unknown
DAINVAPGTSITRTETDQECIDTCKQEDKKCCG–
RSNGVPTCAKICL	Di6.11	2	Unknown
CLAPQRWCSMHDDSLHDDNCCKTCIILWCS	Pu6.20	2	Unknown
CIVGTPCHVCRSQSKSCNGWLGKQRYCGYC	Im9.11	2	Unknown
CCDRPCSIGCVPCCLP	Ca3-VP01, Cp3-VP05	2	Unknown
YWTECCGRIGPHCSRCICPGVVCPKR	Bu25	2	Unknown
WFGHEECTYWLGPCEVDDTCCSASCESKFCGLW	RVIIA	2	Unknown
QCEDVWMPCTSSHWECCSLDCEMYCTQI	Mr6.29	2	Unknown
QCPYCVVHCCPPSYCQASGCRPP	Vc7.4	2	Unknown
QGCCNVPNGCSGRWCRDHAQCC	MIIIA	2	mu
TCSSSSDCPTGQECCPDKLDEPEGSCANECIIT	Pu6.37	2	Unknown
SCSDDWQYCEYPHDCCSWSCDVVCS	Vc6.12	2	Unknown
TCNTPTRYCTLHRHCCSLHCHKTIHACA	Pu6.30	2	Unknown
TTSTRKCKGPLVFCPENHECCSKFCDFIDIPLRYCSTP	Br7.9	2	Unknown
MTKHCTPPEVGCLFAYECCSKICWRPRCYPS	ABVIE	2	Unknown
VCCPFGGCHELCLCCD	MrIIIF	2	Unknown
RCCISPACHDDCICCIT	S3-I05	2	Unknown
RCCISPACHEECYCCQ	S3-Y01	2	Unknown
VSIWFCASRTCSTPADCNPCTCESGVCVDWL	Lt9a variant 2	2	Unknown
QCLPPLSLCTMDDDECCDDCILFLCLVTS	Ar6.5	2	Unknown
STDDCSTAGCKNVPCCEGLVCTGPSQGPVCQPLA	Vn6.18	2	Unknown
GCCDPQWCDAGCYDGCC	Qc3-YDG01	2	Unknown
GCWLCLGPNACCRGSVCHDYCPS	Cal6.4c	2	Unknown
GCSDFGSDCVPATHNCCSGECFGFEDFGLCT	Pu6.25	2	Unknown
STDCNGVPCQFGCCVTINGNDECRELDC	Mr6.23	2	Unknown
RCCTWQECDGNCHCCQ	Cp3-H02	2	Unknown
RCCVHPACHDDCICCIT	Bt3-I03, Vx3-I03	2	Unknown
WWGENDCSWTGPCTVNAECCLGVCDETC	Tx7.31	2	Unknown
GCCHPSTCHVRKGCSRCCS	Tx3g, Vt3-SR01	2	Unknown
SSDEECVGLSGYCGPWNNPPCCSWWECEVYCAVPGPSF	Mi034	2	Unknown
SCCNAGFCRFGCTPCCY	Tx3e, Vt3-TP01, Ec3-TP01-2	2	Unknown
TCDPYYCNDGKVCCPEYPTCGDSTGKLICVRVTD	Im6.7	1	Unknown
TCLEIGEFCGKPMMVGSLCCSPGWCFFICVG	Pc6b	1	Unknown
CGGYSTYCEVDSECCSDNCVRSYCTLF	TxVIIA	1	gamma
GCCCNPACGPNYGCGTSCSRPSEP	S1.7	1	Unknown
TRGCKSKGSFCWNGIECCGGNCFFACVY	Cl6.6b	1	Unknown
CFESWVACESPKRCCSHVCLFVCT	Pn6.6	1	Unknown
WREGSCTSWLATCTDASQCCTGVCYKRAYCALWE	TxMEKL-022/TxMEKL-021	1	Unknown
YCSDSGGWCGLDPELCCNSSCFVLC	Cl6.8	1	Unknown
YCSDDWQPCSHFYDCCKWSCNNGYCP	Vc6.25	1	Unknown
CCDDSECSYSCWPCCY	TxMMSK-02, Cp3-WP03, Vr3-WP04, S3-WP01, Rt3-WP01	1	Unknown
WRVDSECISFWGSCTVDADCCFNSCDETYGYC	Tx7.30	1	Unknown
CCDWPCTIGCVPCCLP	TsMMSK-021	1	Unknown
CCFWPMCRGCDCCYL	Lv3-D02	1	Unknown
CCGPTACLAGCKPCCY	Tx3-KP03	1	Unknown
CESYGKPCGIYNDCCNACDPAKKTCT	Conotoxin-3	1	Unknown
VQPSECKLPAAKGPCKGKYRKVYFNNFKKQCRM–
FTYGGCGGNGNKFRNAKECYHKCAYGV	conkunitzin-G1	1	Unknown
VCCSFGSCDSLCQCCD	Mr3.16	1	Unknown
CCLWPECGGCVCCYL	Lv3-V02	1	Unknown
TRGCKTKGTWCWASRECCLKDCLFVCVY	Cl6.10	1	Unknown
CCSVSICQSPPVCECCA	S3-E03	1	Unknown
CCVVCNAGCSGNCCS	Ts3-SGN01	1	Unknown
SCSGSGYGCKNTPCCAGLTCRGPRQGPICL	Vn6.16	1	Unknown
RCCIWPECGSCVCCL	Cp3-V08	1	Unknown
SCGNLHEMCNYHLPCCRPWRCRASRTGTR–
CLNKPRYRPV	Pu6.13	1	Unknown
RDCRPVGQYCGIPYEHNWRCCSQLCAIICVS	PuIA	1	omega
GCCGSFACRFGCVPCCV	MrIIIA	1	Unknown
GCCHLLACRMGCTPCCW	Tx3-TP01	1	Unknown
GCCIEPLCYQYDCDCCRYL	Cp3-D03	1	Unknown
ECSSPDESCTYHYNCCQLYCNKEENVCLENSPEV	LtVIB	1	Unknown
ECRGYNAPCSAGAPCCSWWTCSTQTSRCF	Vc6.10	1	Unknown
GCCPIGPCMQSVCSPCCP	Vr3-SP01	1	Unknown
GMWGKCKDGLTTCLAPSECCSGNCEQNCKMW	TxMEKL-011, LeD51	1	Unknown
GVWSECSDWLAGCSSPSECCSEKCDTFCRLW	G6.8	1	Unknown
GWDTPAPCRYCQWNGPQCCVYYCSSCNYEEARE–
EGHYVSSHLLERQ	Cal6.3a	1	Unknown
DECCEPQWCDGACDCCS	LtIIIA	1	iota
KFILHALGQWQCCTMQWCDKACYCCE	Vc3.4	1	Unknown
DDCTTYCYGVHCCPPAFKCAASPSCKQT	Cal6.5a	1	Unknown
KTCQRRWDFCPGSLVGVITCCGGLICFLFFCV	Om6.6	1	Unknown
LCPDYTEPCSHAHECCSWNCYNGHCTG	Gla(3)-TxVI	1	Unknown
MQGKISSEQHPMFDPIEGCCTQSCTTCFPCCLI	Lt3.6	1	Unknown
DCCSMSACVPPPACECC	Mi3-E04	1	Unknown
DCCPLPACPFGCNPCCGWPALLSGPHQVMNNE	Mr020	1	Unknown
DCCGVKLEMCHPCLCDNSCKNYGK	PIVE	1	kappa
DAMQKSKGSGSCAYISEPCDILPCCPGLKCNEDFVPICL	LtVIA	1	Unknown
NPKLSKLTKTCDPPGDSCSRWYNHCCSKLCTSR–
NSGPTCSRP	LiCr95	1	Unknown
QCADLGEECYTRFCCPGLRCKDLQVPTCLLA	Ar6.10	1	Unknown
QCCDSNSCEYPKCLCCN	Tx3-L02, Vr3-L01, Vt3-L01, S3-L02	1	Unknown
CVEDGDFCGPGYEECCSGFCLYVCI	Pu6.2	1	Unknown
QKCCGKGMTCPRYFRDNFICGCC	CnIIIG	1	Unknown
QQCCPPVACNMGCEPCC	TxMMSK-04, Vt3-EP01	1	Unknown
RCCGEGASCPVYSRDRLICSCC	CnIIIE	1	Unknown
RCCISPACNDTCYCCQD	Vr3-Y02, Vt3-Y01, Ts3-Y01	1	Unknown
CPNTGELCDVVEQNCCYTYCFIVVCPI	Mr6.2	1	Unknown
RCCTGKKGSCSGRACKNLKCCA	SxIIIA	1	mu
APWTVVTATTNCCGITGPGCLPCRCTQTC	A4.4	1	Unknown

**Table 3 marinedrugs-18-00256-t003:** List of sequences containing eight cysteines in order of interest for experimental characterization, based on the degree (sequence coverage) in alignment graphs (cf. Figure 2). Name or names of sequences are taken from the Conoserver database [40]. Multiple names for the same sequence indicate the same sequence is produced by different species or has different post-translational modifications. Node degree corresponds to the number of sequences with pairwise alignments that are long enough and have a high enough percent identity to be homology modeled with the given sequence as a template. Cysteines are highlighted in red to guide the eye. We note in the fourth column the pharmacological family, although it is unknown for the majority of sequences, as it requires a separate experimental determination in most cases.

Sequence	Name(s)	Degree	Pharm. Fam.
TDVCKKSPGKCIHNGCFCEQDKPQGNCCDSGGC–
TVKWWCPGTKGD	Cal12.1p2	28	Unknown
GHVPCGKDGRKCGYHADCCNCCLSGICKPSTSW–
TGCSTSTVQLTR	R11.10	18	Unknown
QCTPKNQICEEDGECCPNLECKCFTRPDCQSGYKCRP	Vr15b	14	Unknown
CFPPGVYCTRHLPCCRGRCCSGWCRPRCFPRY	Cp1.1	10	Unknown
QCTQQGYGCDETEECCSNLSCKCSGSPLCTSSYCRP	Cap15a	9	Unknown
SCDSEFSSEFCEQPEERICSCSTHVCCHLSSSK–
RDQCMTWNRCLSAQTGN	Gla-MrII, Eu12.4	9	Unknown
SRCFPPGIYCTPYLPCCWGICCGTCRNVCHLRF	Em11.8	8	Unknown
DKWGTCSLLGKGCRHHSDCCWDLCCTGKTCVMT–
VLPCLFLSLIVRWT	Mr11.1	6	Unknown
TCSLPGDGCIRDFHCCGHMCCQGNKCVVTVRRCFNFPY	Pu11.5	6	Unknown
YDAPYCSQEEVRECQDDCSGNAVRDSCLCAYDPAGSP–
ACECRCVEPW	Cal22d	5	Unknown
GTCSGRGQECKHDSDCCGHLCCAGITCQFTYIPCK	Tx11.3	5	Unknown
GTCSYLGEGCKRDSDCCGHFCCGGKTCVITARPCKV	Vc11.4	5	Unknown
RGVCSTPEGSCVHNGCICQNAPCCHPSGCNWANVCPG–
YLWDKN	Cal12.2c	4	Unknown
TCSDLGQACVHESDCCAQMCCLNKKCAMTMPPCNFY	Vc11.1	3	Unknown
CLSEGSPCSMSGSCCHKSCCRSTCTFPCLIP	Ep11.12	2	Unknown
TCSNKGQQCGDDSDCCWHLCCVNNKCAHLILLCNL	M11.2	2	Unknown
RCSDDTGATCSNRFDCCESMCCIGGHCVISTVGCP	Im11.14	1	Unknown
CRLEGSSCRRSYQCCHKSCCIRECKFPCRWV	Vi11.5	1	Unknown
TRSFADLPDDWGMCSDIGEGCGQDYDCCGDMCCDGQI–
CAMTFMACMF	Vc11.6	1	Unknown
CLRDGQSCGYDSDCCRYSCCWGYCDLTCLIN	Im11.1	1	Unknown
CNGRGEWCSTHRSCCDSGDVCCITTPVGPICTRGCSG–
RIIPQRRGAQLRHFF	Pu11.9	1	Unknown
CRAEGTYCENDSQCCLNECCWGGCGHPCRHP	BtX, Sx11.2	1	kappa
CTSEGYSCSSDSNCCKNVCCWNVCESHCRHPGKR	Lt11.3	1	Unknown
CRSGKTCPRVGPDVCCERSDCFCKLVPARPFWRYRCICL	Mr15.2	1	Unknown
DCPTSCPTTCANGWECCKGYPCVRQHCSGCNH	De13b	1	Unknown
EGGYVREDCGSDCMPCGGECCCEPNSCIDGTCHHESSPN	Mi045	1	Unknown
SCRNEGAMCSFGFQCCKKKCCMSHCTDFCRNP	Vt11.3	1	Unknown
WPRLYDSDCVRGRNMHITCFKDQTCGLTVKRNGRLNC–
SLTCSCRRGESCLHGEYIDWDSRGLKVHICPKPWF	Mr22.1	1	Unknown
MCLSLGQRCGRHSNCCGYLCCFYDKCVVTAIGCGHY	Bt11.4	1	Unknown
ASICYGTGGRCTKDKHCCGWLCCGGPSVGCVVSVAPC	Ca11.3	1	Unknown

**Table 4 marinedrugs-18-00256-t004:** List of sequences containing ten cysteines in order of interest for experimental characterization, based on the degree (sequence coverage) in alignment graphs (cf. Figure 2). Name or names of sequences are taken from the Conoserver database [40]. Multiple names for the same sequence indicate the same sequence is produced by different species or has different post-translational modifications. Node degree corresponds to the number of sequences with pairwise alignments that are long enough and have a high enough percent identity to be homology modeled with the given sequence as a template. Cysteines are highlighted in red to guide the eye. We note in the fourth column the pharmacological family, although it is unknown for the majority of sequences, as it requires a separate experimental determination in most cases.

Sequence	Name(s)	Degree	Pharm. Fam.
DRDVQDCQVSTPGSKWGRCCLNRVCGPMCCPAS–
HCYCVYHRGRGHGCSC	Cp20.1	19	Unknown
LHCYEISDLTPWILCSPEPLCGGKGCCAQEVCD–
CSGPACTCPPCL	Lt15.6	5	Unknown
YNRQCCIDKTYDCLKKYRGRENTFASVCQQEAA–
VYCGAWDEAEGCCYGYSHCMSMYAQQSGLDVA–
HNGCKDRKCDNP	Vc21.1	2	Unknown
QCTLVNNCDRNGERACNGDCSCEGQICKCGYRV–
SPGKSGCACTCRNA	Ac8.1	2	Unknown
GCSGTCRRHRDGKCRGTCECSGYSYCRCGDAHH–
FYRGCTCTC	Ca8c	2	Unknown
TCDPTPDCRTTVCETDTGPCCCPHGYNCQTTNS–
GRRACVLVCPHNCPP	Pu19.1	1	Unknown
SGSTCTCFTSTNCQGSCECLSPPGCYCSNNGIR–
QRGCSCTCPGT	G8.3	1	Unknown
GCTRTCGGPKCTGTCTCTNSSKCGCRYNVHPSG–
WGCGCACS	GVIIIA	1	sigma
GCTISCGYEDNRCQGECHCPGKTNCYCTSGHHN–
KGCGCAC	Tx8.1	1	Unknown

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
