# Peer review of "Graph-Directed Approach for Downselecting Toxins for Experimental Structure Determination"

_marinedrugs, 2020, doi:10.3390/md18050256_

Round 1
Reviewer 1 Report
The approach to model structures of cysteine rich small proteins without strong consideration of disulfide bonds is a significant flaw. The utility of these results is unclear. It is not clear how a reader would benefit from or apply these results.
- It is stated at the end of the discussion that "Characterizing multiple possible disulfide isomers is outside the purview of homology modeling". As such the modeling is of no use. A look at some of the structures in the PDB that are in the Tables shows that each contains multiple disulfide bonds. With such small peptides these will be a major determinant of the structure.
- It is not clear how the library that is generated of toxin structures will be used in high-throughput screening. If these toxins are being considered as therapeutic leads, what would they be screened against in silico? They are small proteins. It is not like screening small molecules that will bind to site on a protein.
Reviewer 2 Report
The manuscript reports an interesting approach for prioritizing conotoxins for structural determination and perfectly fits into the scope of the virtual issue. I would like to recommend publication in Marine Drugs, once some minor issues have been addressed:
1) graph-based methods are only briefly introduced. For the broad readership, a bit more information in the introduction might be helpful.
2) When searching the pdb for ‘conotoxin’, the majority of structures are solution NMR and only a small fraction is X-ray (and one EM-structure). The authors should comment on this with a focus on how this might influence the results.
3) The authors show a lot of sequences (Table 1-4). I would suggest to highlight the cysteins and show them in a sequence-aligned way.
4) Figure 4 is really nice, but everything could be a little bit larger.
Reviewer 3 Report
The marine mollusks of the genus Conus are venomous snails that hunt a diverse array of preys by producing a complex venom containing 50-200 different biological peptides known as conotoxins. Typical conotoxins are small (12-30 aa), exhibit a highly constrained structure stabilized by disulfide bridges and post-translational modifications and display high selectivity for many of the cell surface receptors and ion channels found in their prey. Because of these peculiar features, conotoxins are seen as promising natural source for the development of potential therapeutic compounds. Of the approximately 6,000 known conotoxins only 3% of them have associated structural characterization.
In their manuscript titled "Graph-directed approach for downselecting toxins for experimental structure determination" Mansbach R.A. and coll. employed a graph based approach with homology modeling to expand the library of conotoxin structures with the purpose of identifying those members of greatest values for further experimental characterization.
They started by constructing 4 different libraries based on the number of Cys contained in the peptide sequence. By using these library as template and homology modeling allowed to them to expand (approx. 3 times) the library of conotoxin structures usable for High-Throughput Screening.
Taking into account a couple of limitations that the authors did not address in the manuscript, namely post-translational modifications such as the disulfide isomers, the hydoxylated, amidated and carboxylated aminoacids, I appreciated the manuscript.
Below are listed a couple of minor issues that the authors should address prior publication
Minor issues:
1) based on the conotoxins ranking tables (tables 1-4) did the authors noticed any enrichment in terms of conotoxin pharmacological family amongst the top ranking sequences. I mean alpha, gamma, delta or whatever else. This could be of great interest for those readers having more physiological/pharmacological background;
2) pag. 4 lines 114-115: please check the color code. In the manuscript version I have got, I noticed a color mistake;
3) in a recent paper by Younis et al, you referred in the manuscript, they employed a structure-based virtual screening and molecular dynamics simulation assays to assess the therapeutic potential of conotoxins against the lysophosphatidic acid G-protein coupled receptor. For their screening they used 148 conotoxin structures. Conversely, you started with 142. In your screening did you find any additional conotoxin with a structure potentially similar to that of those identified by Younis and coll. Please comment a little bit the issue;
4) based on their data what do the authors think about the hypothesis of using their collection to perform functionally screens on simple models such as non-mammalian organisms.
Round 2
Reviewer 1 Report
The responses from the authors to my previous review did not satisfy my concerns.
Reviewer 2 Report
The authors addressed all points raised in the first round of review.
Reviewer 3 Report
The reviewer thanks very much the authors because, in the present revised version, they properly and extensively addressed the minor issues raised.